# Cytoplasmic Human TDP-43 Mislocalization Induces Widespread Dendritic Spine Loss in Mouse Upper Motor Neurons

**DOI:** 10.3390/brainsci11070883

**Published:** 2021-06-30

**Authors:** Marcus S. Dyer, Adele Woodhouse, Catherine A. Blizzard

**Affiliations:** 1Menzies Institute for Medical Research, College Health and Medicine, University of Tasmania, Hobart, TAS 7000, Australia; Marcus.Dyer@utas.edu.au; 2Wicking Dementia Research and Education Centre, College Health and Medicine, University of Tasmania, Hobart, TAS 7000, Australia; Adele.Woodhouse@utas.edu.au

**Keywords:** amyotrophic lateral sclerosis, TDP-43, dendrite spine

## Abstract

Amyotrophic lateral sclerosis (ALS) is defined by the destruction of upper- and lower motor neurons. Post-mortem, nearly all ALS cases are positive for cytoplasmic aggregates containing the DNA/RNA binding protein TDP-43. Recent studies indicate that this pathogenic mislocalization of TDP-43 may participate in generating hyperexcitability of the upper motor neurons, the earliest detectable change in ALS patients, yet the mechanisms driving this remain unclear. We investigated how mislocalisation of TDP-43 could initiate network dysfunction in ALS. We employed a tetracycline inducible system to express either human wildtype TDP-43 (TDP-43^WT^) or human TDP-43 that cannot enter the nucleus (TDP-43^ΔNLS^) in excitatory neurons (Camk2α promoter), crossed Thy1-YFPH mice to visualize dendritic spines, the major site of excitatory synapses. In comparison to both TDP-43^WT^ and controls, TDP-43^ΔNLS^ drove a robust loss in spine density in all the dendrite regions of the upper motor neurons, most affecting thin spines. This indicates that TDP-43 is involved in the generation of network dysfunction in ALS likely through impacting the formation or durability of excitatory synapses. These findings are relevant to the vast majority of ALS cases, and provides further evidence that upper motor neurons may need to be protected from TDP-43 mediated synaptic excitatory changes early in disease.

## 1. Introduction

Amyotrophic Lateral Sclerosis (ALS) is the most common form of motor neuron disease. ALS was once believed to be primarily neuromuscular and is now characterized by a large neurodegenerative component [1]. ALS involves the progressive loss of upper motor neurons in the motor cortex and lower motor neurons in the spinal cord. This leads to muscle weakness, muscle loss and eventually death in 3–5 years for most patients [2]. ALS is clinically heterogenous, a heterogeneity which can be effectively correlated to degree of upper and lower motor neuron burden [3,4]. Multiple disease mechanisms have been implicated in ALS and it is highly likely to be multifactorial in mechanism [5,6]. Insufficient understanding of the cause of ALS or how pathology progresses through the neuromotor system has hampered the development of effective therapeutics. 

Upper motor neurons play an integral role in ALS pathophysiology. One of the earliest detectable clinical markers of ALS is motor cortex hyperexcitability [7,8]. This increased firing of upper motor neurons within the motor cortex precedes any changes in lower motor neurons [8]. This indicates that the excitability changes may spread through synaptic connections of the corticomotor system network and affect the health of lower motor neurons through glutamate excitotoxicity [9]. Evidence for network dysfunction in ALS can be further demonstrated by intracortical facilitation, a proxy measurement for excitatory synaptic input, which is increased, and short-interval intracortical inhibition (a proxy measurement for inhibitory input), which is decreased [10], suggesting that synaptic mechanisms contribute to altered excitability in ALS. In line with this, studies in models of ALS and post-mortem tissue indicate that synaptic disruption is a conserved feature of ALS [11,12,13,14,15,16]. However, whilst ALS features synaptic dysfunction and the progressive loss of neurons in distinct anatomical networks; the events initiating these pathological events are still unknown.

Familial ALS has been associated with a number of inherited genetic mutations including C9orf72, TDP-43, SOD1 and FUS as some of the most common [17]. However, this familial inheritance of a predisposition to disease only accounts for 10% of total ALS cases. The majority of ALS patients (90%) have no known genetic cause. Despite this, more than 95% of all ALS cases have accumulation of the DNA/RNA binding protein Tar DNA Binding protein of 43kDa (TDP-43) in the cytoplasm of neurons within the corticomotor system [18,19]. This implies that the mislocalization of TDP-43 is critical to the progression of ALS irrespective of mutation status of the person living with ALS. Recent discoveries now link wild-type TDP-43 pathology with hyperexcitability in the motor cortex [20,21]. These studies have provided the bridge between the most frequent pathology of ALS and a conserved pathophysiological presentation of the disease, highlighting the potential importance of TDP-43 mislocalization. However, how these changes commence in people with ALS remains to be fully elucidated.

Therefore, the current study interrogated how TDP-43 mislocalization, the precursor to the main pathology of ALS, affects excitatory synapses in a mouse model of pathogenetically mislocalized TDP-43 through the analysis of dendritic spines. Dendritic spines are small, dynamic protrusions from the shaft of dendrites [22]. The dendritic spine contains the postsynaptic molecular machinery that enables synaptic transmission and plasticity, functionally compartmentalising intracellular chemical and electrical signalling [23]. Neocortical neurons such as upper motor neurons have a highly structured dendritic tree which traverses the laminar of the neocortex, with spine inputs dependent upon the laminar the which spine resides [24,25]. Dendritic spines also exhibit a wide range of sizes and shapes within a single dendritic section, which can be categorized as thin or mushroom/stubby [26]. This regional and structural organization enables the sophisticated fine tuning of neuronal excitability that spine populations are constantly achieving. To investigate if dendritic spines are altered in upper motor neurons with the presence of mislocalized TDP-43, we utilized two inducible mouse models of TDP-43 that express wildtype human (TDP-43^WT^) and mislocalized human TDP-43 (TDP-43^ΔNLS^) in forebrain neurons using the Camk2α promoter [27]. The TDP-43^ΔNLS^ mouse model expresses TDP-43 which accumulates in the cytoplasm and causes endogenous TDP-43 to be lost from the nucleus, as occurs in most cases of ALS [27]. The total levels of TDP-43 in the TDP-43^ΔNLS^ brain are higher than that in TDP-43^WT^, presumably due to impaired regulation of the TDP-43 in the cytoplasm [27]. We analysed these mice at post-natal day 60, with 30 days of induced transgene expression, a timepoint which we have previously shown is before extensive neuron loss [20]. To quantify the density of spines in the primary motor cortex the TDP-43^WT^ and TDP-43^ΔNLS^ mice were crossed with Thy1-YFPH mice, which express fluorescent protein in layer V neurons in the cortex [28]. We analysed different spine morphologies in different compartments of upper motor neurons, which are the main excitatory output for the motor cortex.

## 2. Materials and Methods

### 2.1. Mouse Cohort

All experiments were approved by the Animal Ethics Committee of the University of Tasmania (#A16593) in accordance with the Australian Code of Practice for the Care and Use of Animals for Scientific Purposes (2013). Mice were housed in individually ventilated cage at 20 °C on a 12-h light—dark cycle with ad libitum access to food and water. All mice strains were purchased from Jackson Laboratories (Bar Harbor, USA). Heterozygous *Thy1-YFP* transgenic mice (B6.Cg-Tg(Thy1-YFP)HJrs/J/ Stock No: 003782) [28,29] were crossed with previously described Camk2α-tTa (B6;CBA-Tg(Camk2a-tTA)1Mmay/Stock No: 003010). Mice heterozygous for *Thy1-YFP* and Camk2α-tTa were bred with tetO-TDP-43ΔNLS (B6;C3-Tg(tetO-TARDBP*)4Vle/J/Stock No: 014650) and tetO-TDP-43^WT^ mice (B6;C3-Tg(tetO-TARDBP)12Vle/J/Stock No: 016841) [20,27]. These strains have inducible expression of TDP-43 with the removal of doxycycline from their diet. The breeders were provided chow containing 200 mg/kg doxycycline at least 1 week before mating and their offspring remained on the doxycycline diet before being switched to standard chow at postnatal day 30 (P30). All mice were fully backcrossed to the C57Bl6/J background strain prior to the commencement of experiments. Mono-transgenic Thy1-YFP-H were from a separate mouse cohort and did not deceive doxycycline. Three groups of animals were included in the study, triple transgenic Camk2a-tTA:tetO-TARDBP12:Thy1-YFP-H mice termed TDP-43^WT^, triple transgenic Camk2a-tTA:tetO-TARDBP4:Thy1-YFP-H mice termed TDP-43^ΔNLS^ and mono transgenic Thy1-YFP-H termed WT. All mice were aged P60 at the time of tissue processing, mice of both sexes were used in the current experiments.

### 2.2. Tissue Processing

Mice were terminally anesthetised with an overdose of sodium pentobarbitone (Jurox, Sydney, Australia) delivered with an intraperitoneal injection and transcardially perfused with 4% (*w*/*v*) paraformaldehyde (PFA, Electron Microscopy Sciences, Hatfield, PA, USA) in 0.01 M phosphate-buffered saline (PBS, Sigma Aldrich, St Louis, MO, USA). Brains were removed and post-fixed in 4% PFA overnight at 4 °C and stored in PBS containing 0.1% sodium azide.

Brains were dissected and cut on the coronal axis at bregma −4.00 mm. The rostral segment was embedded in 5% agarose for sectioning. Then, 40 µm coronal sections were sectioned using a Leica VT100S vibratome (Wetzlar, Germany). Free-floating sections were collected and stored in 24 well plates containing 0.1% sodium azide in PBS and stored at 4 °C until required for immunohistochemistry.

### 2.3. Immunohistochemistry

Coronal sections containing primary motor cortex [30] were collected from each animal. Free floating sections were first washed twice for ten minutes in 0.01 M PBS before a blocking solution containing 1% normal goat serum and 0.3% triton-X (Sigma Aldrich USA) in 0.01 M PBS was added for one hour. Sections were incubated with anti-GFP antibody (1:2000, Nacalai Tesque, Cat# 04404-84, Kyoto, Japan), anti-hTDP-43 antibody (1:1000 Proteintech Cat# 60019-2-Ig, Rosemont, IL, USA) diluted in 0.3% Triton-X in 0.01 M PBS. Sections were washed three times for ten minutes in PBS before being incubated with anti-rat AlexaFluor 488 secondary antibody, anti-mouse AlexaFluor 568 secondary antibody and DAPI (Invitrogen, Boston, MA, USA) for 90 min. All incubations were performed at 21 °C. Sections were mounted onto slides and coverslipped with PermaFluor (Boston, MA, USA) aqueous mounting media.

### 2.4. Microscopy

Immunofluorescent images were captured using an UltraView Nikon Ti spinning disk confocal microscope with Volocity software (Perkin Elmer, Boston, MA, USA). A low magnification objective (20×) was used to determine the location of the motor cortex. For TDP-43 localisation, the objective was switched to a higher magnification (40×, N/A 0.95) objective and images with z-spacing of 1 µm were collected. For spine quantification the objective was switched to a high magnification (60× N/A 1.2) water immersion objective and images with z-spacing of 0.5 µm were collected. Standard excitation and emission filters were used. Images of basal dendrites attached to the cell body were taken of pyramidal neurons in layer V of the primary motor cortex (Figure 1). The objective was moved to layer II/III, identified through an increased density of nuclei between the YFP cell bodies of layer V and layer I, and images were taken of the apical dendrites (Figure 1). The objective was then moved to layer I, identified due to its proximity to the surface and images were taken of apical tuft dendrites (Figure 1). Three-dimensional image rendering with volume and surface rendering added was performed in Imaris 9.2.0 (Zurich, Switzerland). 

### 2.5. Quantification

Dendritic spine density quantification was performed using Neurolucida software (MBF Bioscience, Burlington, MA, USA). Individual basal, apical and apical tuft dendrites were traced through z-stacks and their spines labelled as either thin or mushroom spines depending on the spine head size [31]. Stubby spines were classed as mushroom shaped, and filopodial spines were classed as thin spines. Data files were then exported to Neurolucida explorer 11 (MBF Bioscience, Burlington, MA USA) for quantification. Branched structure analysis was used to determine the density and ratio of each spine type. Basal dendrites were analysed according to branch order, with branch order 1 dendrites being directly attached to a layer V pyramidal cell soma. Branch order 1 dendrites were excluded from the analysis as these had variable length but very few spines, basal dendrite analysis included branch order 2 and above. 

### 2.6. Statistical Analysis

Basal dendrite data of n = 5 for WT and n = 3 for TDP-43^WT^ and TDP-43^ΔNLS^ were summed for each separate mouse and using G*Power [32] it was determined that these mouse numbers were sufficiently powered for the current dataset and we commenced the rest of the analysis. Data was analysed per dendrite using GraphPad Prism version 9 (San Diego, CA, USA). All data is represented as the mean ± standard error of the mean, each datapoint represents an individual dendrite. Data was checked for normality and where the data sets passed the normal distribution assumption (majority), one way analysis of variance (ANOVA) with a post hoc Bonferroni correction for multiple comparisons was used. In the apical data set the TDP-43^ΔNLS^ group did not demonstrate sufficient evidence for normal distribution assumption and the Kruskal–Wallis one-way ANOVA was used for this data set with a Dunn’s multiple comparison. A *p*-value of <0.05 was considered statistically significant.

## 3. Results

Here, we utilise two inducible mouse models of TDP-43 expression. The TDP-43^WT^ and TDP-43^ΔNLS^ mouse models express hTDP-43 in forebrain neurons utilising the Camk2α promoter. TDP-43^WT^ mice express full length human TDP-43 with no mutation, whereas TDP-43^ΔNLS^ mice express TDP-43 with a mutated nuclear localisation sequence, causing TDP-43 to accumulate in the cytoplasm. Expression in these models is controlled by the presence of doxycycline in the diet of the mice. For these experiments, doxycycline was removed from the diet at post-natal day 30 (P30), inducing expression of the TDP-43 transgene for 30 days until experiments occurred at P60. These mice were crossed with Thy1-YFPH mice, which express fluorescent protein in layer V neurons in the cortex. Using immunohistochemistry directed that YFP and human TDP-43, we identified that human TDP-43 expression was restricted to the nucleus in the TDP-43^WT^ mice of all YFP cells visualized (Figure 2B), whereas in TDP-43^ΔNLS^ mice human TDP-43 was predominantly cytoplasmic and excluded from the nucleus in all YFP positive cells visualized in this model (Figure 2C,D).

### 3.1. Expression of Mislocalised TDP-43 Caused a Loss of Spines in Apical Tuft Dendrites

Neurons receive highly compartmentalised synaptic inputs. Apical tuft dendrites of layer V pyramidal neurons branch off the thick apical dendrite in layer I. The synapses on apical tufts of upper motor neurons are electrically isolated from the cell body and receive a broad range of inputs from long range cortical connections and feedback from the thalamus [33,34]. The density of spines was measured in apical tuft dendrites in WT, TDP-43^WT^ and TDP-43^ΔNLS^ mice (Figure 3A). The total density of spines in apical tuft dendrites was reduced in TDP-43^ΔNLS^ neurons by >44% compared to both WT and TDP-43^WT^ neurons (Figure 3B). The frequency of spine density further shows clear alterations between TDP-43^ΔNLS^ and the WT and TDP-43^WT^ controls with a 49% change in median density (Figure 3C). Overall, this data indicates that TDP-43^ΔNLS^ expression is affecting total spine density in the apical tuft dendrites of Layer V pyramidal neurons of the motor cortex. The characterisation of spines as either thin or mushroom has important synaptic physiological implications. Mushroom spines are a more mature type, generally having a greater number of α-amino-3-hydroxy-5-methyl-4-isoxazolepropionic acid receptor complex (AMPA) and N-methyl-D-aspartate (NMDA) receptors [35]. This means that synapses onto mushroom spines are more readily able to depolarise the membrane than synapses onto thin type spines [36]. When the morphology of spines was assessed, a decrease in mushroom spine density of >29% was detected in TDP-43^ΔNLS^ apical tuft dendrites (Figure 3D), while thin spines were also reduced in TDP-43^ΔNLS^ apical tuft dendrites by 54% (Figure 3E). The greater loss of thin spines caused a reduction in the proportion of thin spines in apical tuft dendrites in TDP-43^ΔNLS^ mice (Figure 3F). This indicates that mislocalised TDP-43 is driving a dramatic decrease in the thin, immature spines of apical tuft dendrites.

### 3.2. The Density of Spines Is Reduced in Apical Trunk Dendrites in TDP-43^ΔNLS^ Neurons

Apical trunk dendrites were identified as the thick, projecting dendrites of layer V pyramidal neurons in the motor cortex (Figure 1). Apical trunk dendrites receive inputs from local circuits and layer IV neurons of the motor cortex [24]. Images were taken in the layer II/III region of these dendrites and the density of spines was determined in WT, TDP-43^WT^ and TDP-43^ΔNLS^ neurons. The total density of spines in apical trunk dendrites was reduced by >44% in TDP-43^ΔNLS^ neurons compared to WT and TDP-43^WT^ neurons (Figure 4B). The frequency distribution of spine density shows a robust leftward shift (loss) of spine density in the apical trunk dendrites of TDP-43^ΔNLS^ neurons compared to WT and TDP-43^WT^ neurons (Figure 4C). There was no significant difference in total spine density between WT and TDP-43^WT^ neurons. When spine type was examined, >31% of mushroom spines were lost on the apical trunk of in TDP-43^ΔNLS^ neurons compared to both WT and TDP-43^WT^ neurons (Figure 4D), while thin spines were decreased by almost 50% (Figure 4E). Because of this increased loss of thin spines, there was a corresponding decrease in the proportion of thin spines that were observed on apical trunk dendrites of TDP-43^ΔNLS^ neurons (Figure 4F).

### 3.3. The Density of Spines Is Reduced in Basal Dendrites in TDP-43^ΔNLS^ Neurons

Basal dendrites were identified as protruding from a layer V pyramidal neuron soma and spines were counted from an entire visible dendritic tree stemming from a single somatic branch point (Figure 5A). Basal dendrites of layer V pyramidal neurons in the motor cortex primarily receive inputs from layer IV neurons and local circuits [24]. The density of spines in basal dendrites was reduced in TDP-43^ΔNLS^ neurons by >55% compared to both WT and TDP-43^WT^ neurons (Figure 5B). The frequency distribution of spines on the basal dendrites of TDP-43^ΔNLS^ neurons shows a left shift in spine density (Figure 5C). The density of mushroom and thin spines on basal dendrites were reduced by >40% and >65%, respectively, in TDP-43^ΔNLS^ neurons (Figure 5D,E). Once again, the exacerbated loss of thin spines meant that there was an increase in the proportion of mushroom spines in basal neurons (Figure 5F).

### 3.4. Changes to Dendritic Spines Were Dependent on the Localisation of TDP-43 but Not the Location of the Dendritic Spine

Regional analysis of dendritic spine density following 30 days of expression of TDP-43^ΔNLS^, in comparison to TDP-43^WT^ expression and wild-type controls, revealed a dramatic loss of spines throughout the dendritic tree of the upper motor neurons. There were no significant differences in the proportion of total, mushroom or thin spine densities compared to WT in TDP-43^WT^ or TDP-43^ΔNLS^ layer V excitatory neurons between the basal, apical or apical tuft regions (Figure 6A). Together, these data suggest that with wild-type TDP-43, it is the mislocalisation of TDP-43 to the cytoplasm which is important for spineopathy in ALS (Figure 6B).

## 4. Discussion

ALS is a currently incurable disease that causes selective degeneration of the corticomotor system. It remains to be elucidated why motor neurons, which are some of the largest in the nervous system, are selectively vulnerable in this disease [37]. However, it is highly likely that, similar to many neurodegenerative diseases, a significant period of neuronal dysfunction occurs prior to frank cell loss, contributing to both the initiation and propagation of neuronal death in the corticomotor system [38,39]. Whilst how neuronal dysfunction arises in ALS is not known, it is becoming apparent that upper motor neuron dysfunction is a critical factor in ALS [2]. The most likely disease profile of an ALS patient is no known genetic mutation, focal motor cortex hyperexcitability which has the propensity to cause widespread network dysfunction and TDP-43 protein which is excluded from the motor neuron nucleus [10,19]. How these critical disease profiles interact is only starting to be comprehended [13,20] and how these changes result ultimately in neuronal demise remains an important research question. Therefore, we endeavoured to interrogate whether TDP-43 mislocalization contributes to the generation of upper motor neuron dysfunction through altering neuronal circuitry. We investigated the role of mislocalised TDP-43 at the dendritic spine by employing an inducible model, in which TDP-43 that cannot enter the nucleus is expressed in layer V pyramidal neurons (on the CamkIIα promoter) within the motor cortex; crossed with the Thy-1 YFP-H mouse to visualize dendritic spines. 

Excitatory synapses can be elegantly investigated at the dendritic spine—the structural and functional representation of the vast majority of excitatory synapses in the brain [36]. Spines are the site for synaptic transmission and plasticity, functionally compartmentalising intracellular chemical and electrical signalling [40]. Spine loss and dendrite pathology is observed in upper motor neurons of ALS patient post-mortem tissue [16,41]. This spine loss has been replicated in preclinical models and shown to be a consistent feature of familial mouse models of ALS at symptomatic time points [42]. The most widely characterized model of familial ALS, the mutant SOD1^G93A^ mouse model, demonstrates progressive spine loss [43]. This is also replicated in the mutant TDP-43^A315T^ model of ALS [14]. Whilst these contributions highlight the importance of spine changes in ALS, it is unclear if they are specific to the mutant protein and hence associated with the relevant familial form of ALS. Here, we determined that TDP-43 mislocalisation could drive changes in spine density, addressing the underrepresented non-familial ALS cohorts in past studies. We discovered that following 30 days expression of mislocalized TDP-43 there is a significant loss of spines in upper motor neurons in comparison to both controls and wildtype human TDP-43 expressing mice. This data validates the functional loss of spontaneous- and mini-excitatory post-synaptic currents previously reported in TDP-43^NLS^ layer V cortical motor neurons in the same model at the same time points [20]. This is consistent with other mouse models of familial ALS, namely SOD1 and FUS, which also have a loss of dendritic spines, but lack TDP-43 pathology [19,44,45]. Furthermore, spine loss has been observed with TDP-43 mutation, where this form of TDP-43 is also present outside the nucleus [14,46]. This suggests that spine loss is a critical step in the progression of ALS [19]. Previously, we have shown that there is no neuron loss in the motor cortex in the TDP-43^WT^ or TDP-43^ΔNLS^ mouse models [20], suggesting that dendritic spines changes occur prior to neuron death. We quantified distinct anatomical regions and observed that dendritic spine loss was widespread, and not restricted to a specific neuronal compartment. Thus, both long-range and local synaptic inputs are uniformly lost in upper motor neurons in TDP-43^ΔNLS^ mice.

In ALS, neuronal loss in the motor cortex is observed post-mortem [16,47] yet the degree of TDP-43 aggregation may not reflect the degree of cell loss [48]. This may be because the neurons that accumulate TDP-43 in the cytoplasm selectively degenerate and die over the course of the disease. Furthermore, the extent of TDP-43 mislocalisation in the motor cortex is controversial [49,50]. For instance, antibodies commonly used in human studies that target only pTDP-43 show only protein aggregation, not the nuclear or cytoplasmic TDP-43 that is soluble [48]. Further studies are necessary to determine if mislocalisation of soluble TDP-43 is having a pathogenic effect prior to and potentially independent of TDP-43 aggregation. TDP-43 localises to synaptic vesicles both under normal conditions [51,52] and in the anterior horn of the spinal cords of human ALS patients [53]. Here, we show that upper motor neuron spine alterations occur when TDP-43 is mislocalised to the cytoplasm, however, the TDP-43^ΔNLS^ mouse does have increased cortical levels of TDP-43 than TDP-43^WT^, so the increased load of TDP-43 in TDP-43^ΔNLS^ could also contribute to spine alterations.

Increases in TDP-43 within the cytoplasmic compartment affects spine density, an effect that is more pronounced in thin spines. Whilst the functional relevance of morphological spine types is unclear [54], evidence suggests that thin spine types are immature, transient and highly motile [55]. This immaturity and motility is critical for the establishment of new connections, which through the recruitment of more synaptic machinery, namely the AMPA receptors, can then develop into the more mature, dense, mushroom spines [56]. Previous research has identified that TDP-43 can transcriptionally regulate components of AMPA receptors [57,58] and that the pathogenic mislocalisation of TDP-43 in this model decreases the expression of the AMPA receptor subunits Gria 1–3 [20]. This could potentially affect the formation of mushroom spines and the maturation of thin spines. Future studies investigating spine plasticity and maturation are warranted to answer these research questions.

Upper motor neuron hyperexcitability is the first physiological change to be detected in ALS patients [3,39] and has been hypothesised to underlie the spread of pathology from the upper motor neurons to the lower motor neurons in the spinal cord; termed the ‘die forward hypothesis’ [9,59]. Changes in neuronal excitability in ALS are measured using transcranial magnetic stimulation (TMS). TMS produces electrical currents to stimulate the primary motor cortex and muscle response can then be measured. Using TMS, studies have shown patients in the early stages of ALS require a smaller stimulus to generate a motor response, indicating hyperexcitable upper motor neurons [8,60]. Preclinical research has offered insight into the mechanisms of the hyperexcitable motor cortex in ALS. Direct electrophysiological recordings can be made from upper motor neurons expressing different disease-causing proteins. Previous studies in the SOD1^G93A^ mouse model at certain time points [13,61], the TDP-43^A315T^ mouse model [15] and the current TDP-43^ΔNLS^ model [20] have all reported that layer V pyramidal neurons in the motor cortex are hyperexcitable, consistent with clinical findings. The relationship between layer V cortical motor neuron hyperexcitability and dendritic spine loss is not yet known, but it is highly likely that homeostatic mechanisms are at play. Thus, it is possible that the spine loss identified in the current study, and in other models, may contribute to the initiation and progression of increased upper motor neuron excitability. To perform useful information processing neurons must integrate a range of inputs to an appropriate output. When synaptic inputs dramatically change, intrinsic excitability is homeostatically regulated to allow suitable responsiveness to synaptic inputs. Previous studies using whisker trimming and auditory deprivation to generate loss of synaptic input have shown this decreased synaptic input can drive increases in intrinsic excitability to maintain functional output [62,63,64]. Therefore, it is possible that spine loss may result in increased intrinsic excitability, through an overcompensation leading to the hyperexcitability which has been observed in the TDP-43^ΔNLS^ motor cortex at the same timepoint as the current experiments [20]. However, it is also plausible that the converse may be happening, that increased intrinsic excitability occurs first leading to a maladaptive down regulation of spine formation [65,66]. Irrespective of whether synaptic or intrinsic excitability changes occur first, we hypothesise that such dramatic changes in excitatory synaptic inputs (~50% reduction) or intrinsic excitability [20] result in homeostatic changes aimed at maintaining neuronal output; but that homeostatic failure ultimately leads to a hypersensitivity to excitatory synaptic inputs. Such failures of the homeostatic control systems could occur via a maladaptive feedback response to perturbations, altered set-point regulation or sensor impairment [67]. Finally, it is also feasible that TDP-43 mislocalisation may mediate direct independent effects on intrinsic excitability and dendritic spines due to the role of TDP-43 in splicing [50]. Regardless of whether spine loss is the cause or consequence of intrinsic excitability changes, spine loss in upper motor neurons is a critical and early feature of ALS progression in models of ALS. 

## 5. Conclusions

We have identified that TDP-43 mislocalization causes a loss of 44–50% of dendritic spines in upper motor neurons regardless of whether these spines were located on basal, apical or apical tuft dendrites. Furthermore, there was a more pronounced loss of thin spines compared to mushroom spines across all dendrite regions. Collectively, these data show that TDP-43 mislocalisation, the most common pathological hallmark in ALS, can cause spine loss and potentially inhibit the formation of new spines in upper motor neurons in ALS. Future therapeutics could look at targeting either the dendritic spine, or the spine in combination with mislocalisation of TDP-43 in upper motor neurons.

## Figures and Tables

**Figure 1 brainsci-11-00883-f001:**
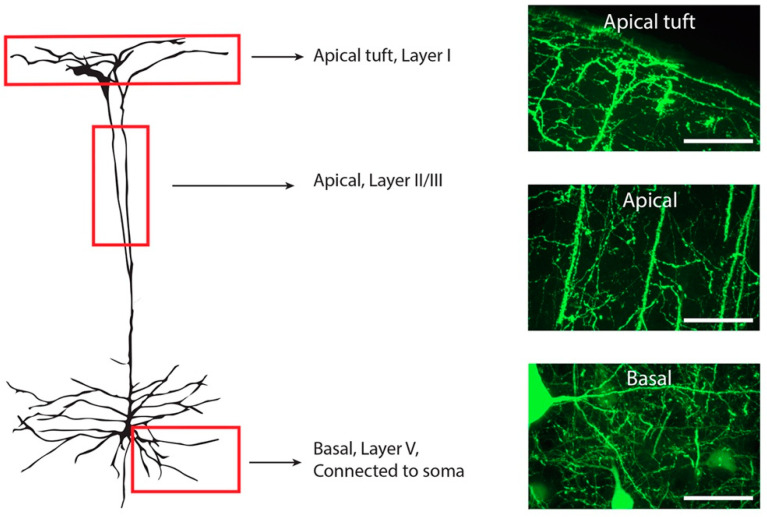
Illustration of a Layer V YFP-H positive neuron within the motor cortex. Three different regions were included for spine quantification (red boxes). All apical tuft dendrites were quantified in layer I, identified by the clearly visible meninges membrane. Apical dendrites within cortical layer II/III were identified through location of the cortical layers and only the main apical dendrite included in quantification. Basal dendrites in layer V were identified as directly connected to a YFP-H pyramidal cell soma. Dendritic spines on the entire visible dendritic tree were counted from a single somatic branch point. Scale bars 50 µM.

**Figure 2 brainsci-11-00883-f002:**
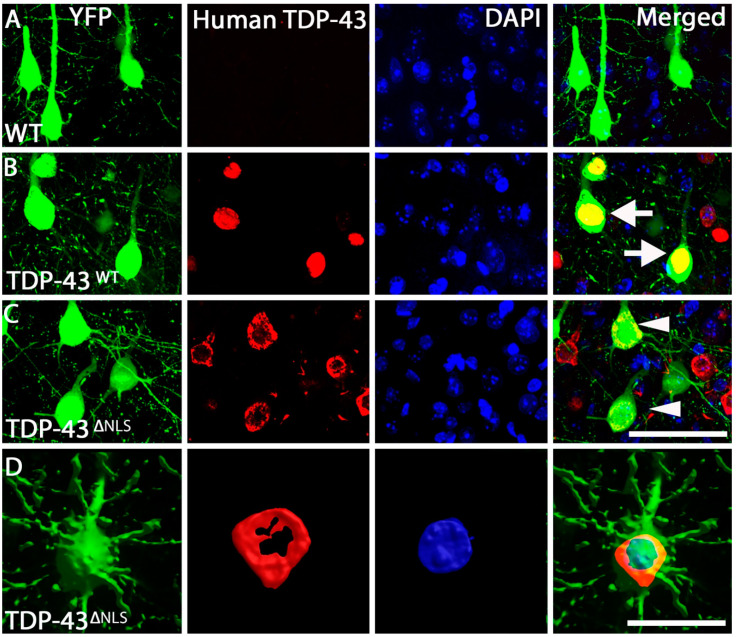
TDP-43 localization within the motor cortex. (**A**). Example of WT Layer V neurons within the motor cortex that are positive for YFP (green) and DAPI (blue), with no human TDP-43 (red) labelling present. (**B**). TDP-43^WT^ Layer V neurons within the motor cortex were positive for YFP (green), human TDP-43 (red) and DAPI (blue), with human TDP-43 labelling nuclear (arrows) (merged). (**C**). TDP-43^ΔNLS^ Layer V neurons within the motor cortex were positive for YFP (green), human TDP-43 (red) and DAPI (blue), with human TDP-43 labelling excluded from the nucleus (arrowheads) (merged). (**D**). Three-dimensional representation of a YFP neuron (green) with TDP-43 (red) wrapping around the DAPI nucleus (blue). Scale bar, A-C = 50 µm, D = 20 µm.

**Figure 3 brainsci-11-00883-f003:**
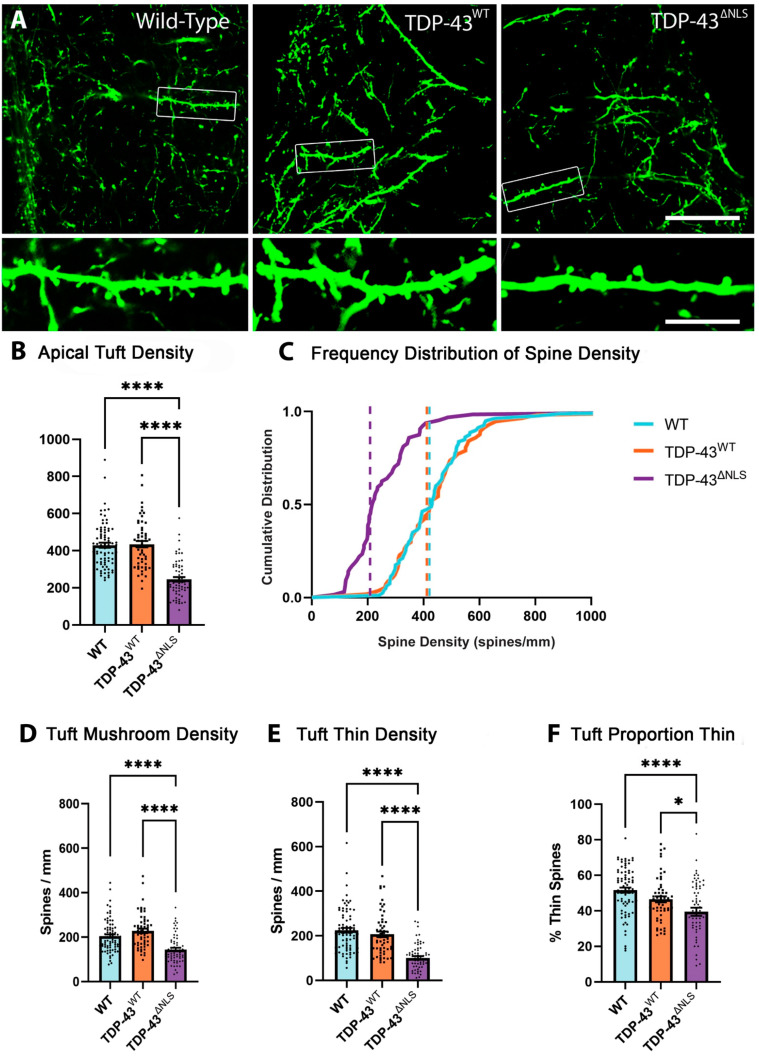
Spines are lost in apical tuft dendrites in TDP-43^ΔNLS^ cortical motor neurons. (**A**). Example images of apical tuft dendrites from WT, TDP-43^WT^ and TDP-43^ΔNLS^ layer V pyramidal neurons of the motor cortex that were quantified. (**B**). The density spines on apical tuft dendrites was reduced in TDP-43^ΔNLS^ mice compared to WT (F_(2, 195)_ = 53.3, *p* < 0.0001) and TDP-43^WT^ mice (*p* < 0.0001). (**C**). Frequency distribution of spine density, dashed lines show median density. Almost all TDP-43^ΔNLS^ apical tuft dendrites have a density lower than the median WT spine density in apical tuft dendrites. (**D**). The density of mushroom spines was reduced in TDP-43^ΔNLS^ apical tuft dendrites compared to WT (F_(2, 195)_ = 22.6, *p* < 0.0001) and TDP-43^WT^ mice (*p* < 0.0001).(**E**). The density of thin spines was reduced in TDP-43^ΔNLS^ apical tuft dendrites compared to WT (F_(2, 195)_ = 38.2, *p* < 0.0001) and TDP-43^WT^ mice (*p* < 0.0001). (**F**) The proportion of thin spines was reduced in TDP-43^ΔNLS^ apical tuft dendrites compared to WT (F_(2, 195)_ = 11.84, *p* < 0.0001) and TDP-43^WT^ (*p* = 0.0326). * *p* < 0.05, **** *p* < 0.0001, one-way ANOVA with Bonferroni multiple comparisons. Scale bar = 30 µm, inset 7.5 µm.

**Figure 4 brainsci-11-00883-f004:**
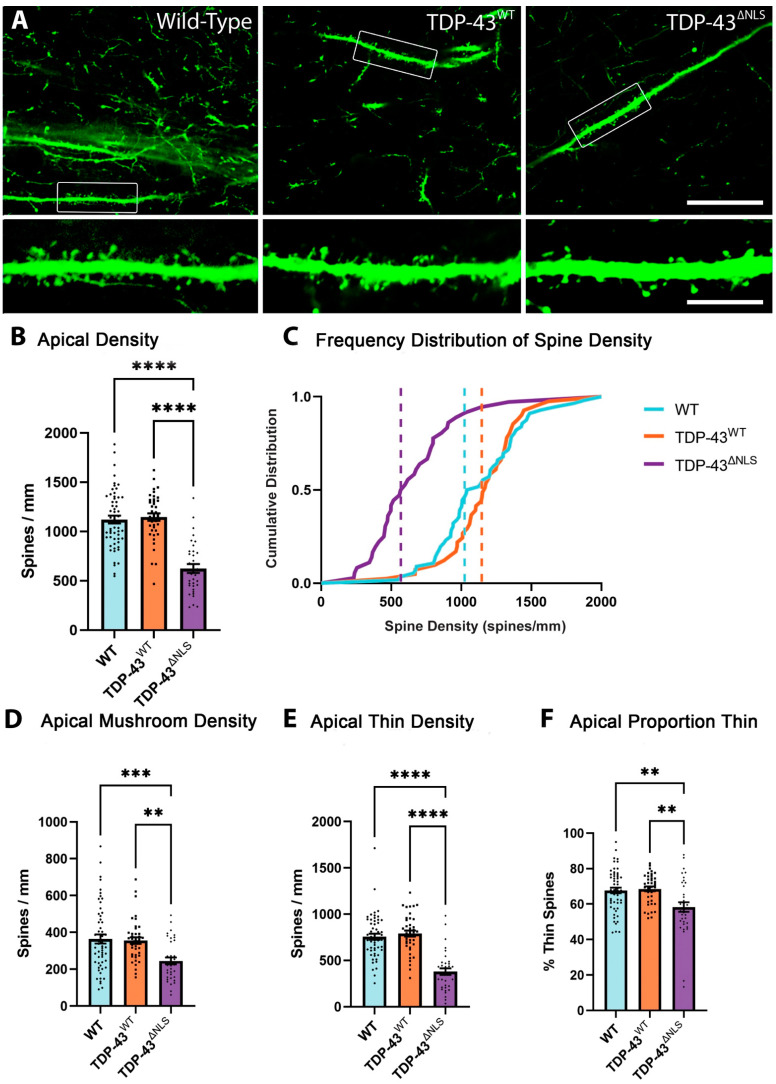
Apical trunk dendrites show reduced spine density in TDP-43^ΔNLS^ layer V cortical motor neurons. (**A**). Representative images of WT, TDP-43^WT^ and TDP-43^ΔNLS^ apical trunk dendrites. (**B**). The density of spines on apical trunk dendrites was reduced in TDP-43^ΔNLS^ neurons compared to WT (H_(2)_ = 51.0, *p* < 0.0001) and TDP-43^WT^ (*p* < 0.0001). (**C**). The frequency distribution of apical trunk spine density, dotted lines represent the median. Almost all WT and TDP-43^WT^ dendrites have a higher density of spines than 50% of TDP-43^ΔNLS^ dendrites. (**D**). There was a reduction in mushroom spine density on TDP-43^ΔNLS^ apical trunk dendrites compared to WT (H_(2)_ = 15.4, *p* = 0.0007) and TDP-43^WT^ (*p* = 0.0042). (**E**). TDP-43^ΔNLS^ apical trunk dendrites exhibit a reduced density of thin spines compared to WT (H_(2)_ = 49.4, *p* < 0.0001) and TDP-43^WT^ (*p* < 0.0001). (**F**). Apical trunk dendrites have a reduced proportion of thin spines in TDP-43^ΔNLS^ neurons (H_(2)_ = 11.6, *p* = 0.0019 to WT and *p* = 0.0015 to TDP-43^WT^). ** *p* < 0.01, *** *p* < 0.001, **** *p* < 0.0001, Kruskal–Wallis one-way ANOVA with Dunn’s multiple comparisons. Scale bar = 30 µm, inset 7.5 µm.

**Figure 5 brainsci-11-00883-f005:**
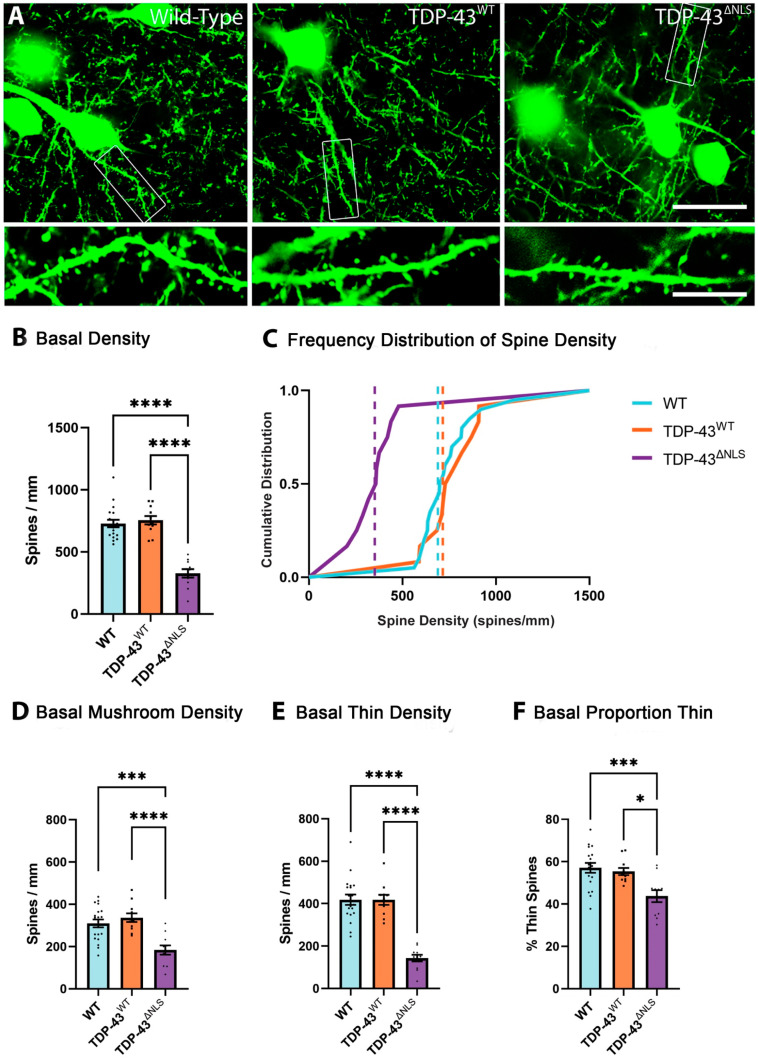
TDP-43^ΔNLS^ layer V pyramidal motor neuron basal dendrites have reduced spine density. (**A**). Example images of basal dendrites from WT, TDP-43^WT^ and TDP-43^ΔNLS^ layer V pyramidal neurons of the motor cortex (scale bar 30um, inset 7.5 um). (**B**). Spine density was reduced in TDP-43^ΔNLS^ basal dendrites compared to both WT and TDP-43^WT^ neurons (F_(2,38)_ = 45.9, *p* < 0.0001). (**C**). Frequency distribution of spine density shows significant leftward shift (decreased spine density) of TDP-43^ΔNLS^ basal dendrites. Dotted lines show the median. (**D**). There was a reduced density of mushroom spines in TDP-43^ΔNLS^ basal dendrites compared to WT (F_(2,38)_ = 13.3, *p* = 0.0003) and TDP-43^WT^ (*p* < 0.0001). (**E**). TDP-43^ΔNLS^ basal dendrites have reduced density of thin spines to both WT and TDP-43^WT^ neurons (F_(2,38)_ = 40.1, *p* < 0.0001). (**F**). TDP-43^ΔNLS^ basal dendrites have a reduced proportion of thin spines (F = 8.57_(2,38)_, *p* = 0.0105 to WT and *p* = 0.0009 to TDP-43^WT^). * *p* < 0.05, *** *p* < 0.001, **** *p* < 0.0001, one-way ANOVA with Bonferroni multiple comparisons.

**Figure 6 brainsci-11-00883-f006:**
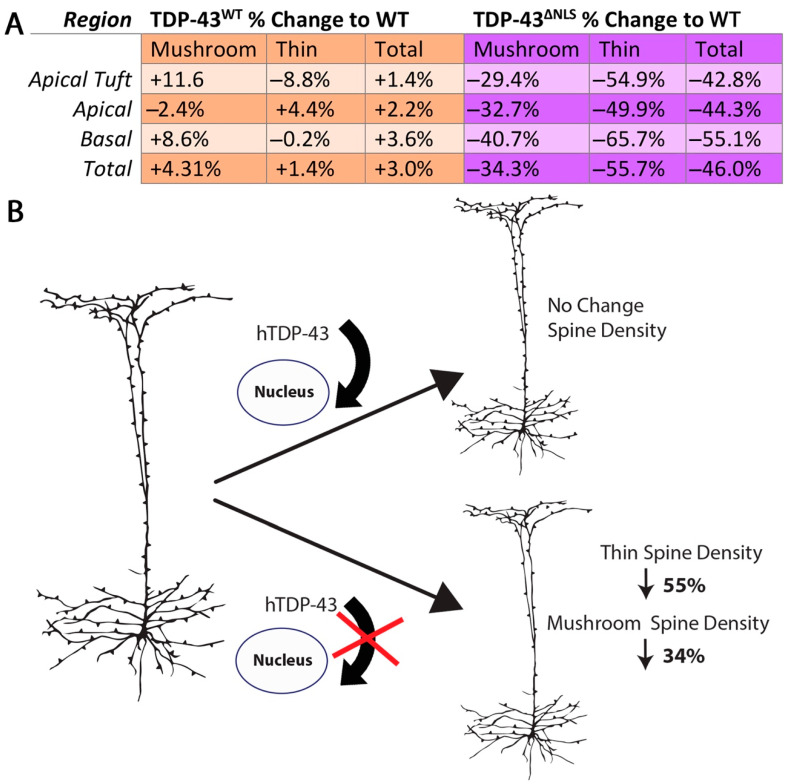
Importance of TDP-43 localisation on dendritic spine density. (**A**). Table showing the percentage difference for mushroom, thin and total spines compared to wild-type of TDP-43^WT^ and TDP-43^ΔNLS^ dendrites (One-way ANOVA with Bonferroni multiple comparisons - Mushroom F_(2,107)_ = 0.79, *p* = 0.454, Thin F_(2,107)_ = 1.20, *p* = 0.303, Total F_(2,107)_ = 1.37, *p* = 0.258) (**B**). Illustration highlighting the importance of TDP-43 localisation on spine density in the upper motor neuron. In TDP-43^ΔNLS^ mice, there was a robust loss of spines in apical tuft, apical and basal dendrites. This loss of spines occurred in both mushroom and thin spines, where thin spines were more severely affected. Overexpression of wild-type TDP-43 did not cause any significant spine changes in upper motor neurons suggesting that it is the localisation, rather than the expression levels of TDP-43 which are important for spineopathy in ALS.

## Data Availability

The datasets generated in the present study are available from the corresponding author upon reasonable request.

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
