# Peer review of "Cytoplasmic Human TDP-43 Mislocalization Induces Widespread Dendritic Spine Loss in Mouse Upper Motor Neurons"

_brainsci, 2021, doi:10.3390/brainsci11070883_

Round 1
Reviewer 1 Report
The manuscript is well written, but please provide detailed stats including F score, t score, degree of freedom, effect size.. etc
Author Response
We thank the reviewer for their time. We have now included the F score, degrees of freedom and p-values in the appropriate figure legends. We have highlighted these changes in red in text.
Reviewer 2 Report
This is a well-presented paper with clear structure and methodology, in which the authors find that transgenic mice with cytoplasmic TDP-43 accumulation show widespread loss of dendritic spines (particularly thin spines). Although the observation linking dendritic spine loss to the presence of cytoplasmic TDP-43 is of interest and may be of value, I have some comments that largely concern wider relevance and implications of these specific findings to ALS disease pathology.
- In prior studies by this group and others, TDP-43 levels are noted to be higher in the TDP-43NLS mouse forebrain than in the TDP-43WT . Did the authors look at this, and can this be of consequence to the findings?
- Did the authors explore the functional relevance of their findings (to the development of disease)? That is, does the loss of dendritic spines (in the presence of cytoplasmic TDP-43) cause cellular hyperexcitability, or neuronal cell death? As this is an induced model of cytoplasmic TDP-43, this is not necessarily a definitive assumption that can be made.
- Other preclinical familial mouse models (SOD1 and FUS) have similarly found loss of dendritic spine density in cortical neurons (as is mentioned by the authors) but these genotypes are notable for their lack of TDP-43 aggregation in vivo. Can the authors comment on this in light of their conclusions? This discrepancy should be discussed.
- Section 3.4 is not clear to me (specifically, the long sentence 275-278 is hard to decipher). The same group (Jiang et al 2019) have shown in previous work that an over-expression of hTDP-43 (TDP-43A315T) caused loss of dendrite spine density in cortical neurons (and this was predominantly nuclear TDP-43 overexpression). How does this concur with the conclusion given here (line 286: “overexpression of TDP-43WT did not cause any significant spine changes in UMN suggesting that it is the localisation, rather than the expression levels of TDP-43, which are important for spineopathy in ALS”)?
There are also some broader points that are worth consideration, particularly in relation to the translatability of these findings to human ALS pathology:
- A prominent feature of TDP-43 cytopathology post-mortem is its distinct aggregation in the cytoplasm. Is aggregation a specific feature of this transgenic cytoplasmic (TDP-43NLS) mouse model? If not, can the authors also comment on how translatable this model may be as well as implications on underlying neurodegeneration (i.e., is neurodegeneration dependent on cytoplasmic aggregation?).
- Conversely, it has been suggested that cytoplasmic TDP-43 inclusions are relatively rare in cortical neurons (specifically pyramidal Betz cells), despite loss of native nuclear TDP-43 and significant neuronal loss. How do the authors square this with the relevance of their TDP-43NLS model and the neurodegenerative hypothesis?
- As this study primarily explores the concept of TDP-43 mislocalisation pathology in ALS (and in relation to the point above), it would be important to mention/discuss the known inconsistency between TDP-43 dysfunction and neuronal loss in vivo, which greatly obscures how this pathology actually contributes to neurodegeneration. This also emphasises the value of linking study findings to evidence of disease pathology.
- Line 52-56: The division into ‘sporadic’ vs ‘familial’ is an increasingly false dichotomy and this terminology should be avoided for a number of reasons, including the lack of clinical or pathological differences between these groups. In particular, the wording of line 55 (“The majority of ALS patients have no known genetic cause, termed sporadic disease..”) is not quite accurate: ‘sporadic’ disease is historically defined by the absence of a (detectable) family history, not the absence of a causative mutation; also, there are more ‘apparently sporadic’ patients who are found to have a gene mutation than there are familial patients with a known mutation. Perhaps the more interesting point on genotypes is that TDP-43 mislocalisation is the most common finding overall but this feature is not shared by all causative genotypes, notably absent in patients with SOD1 and FUS
Minor points:
- hTDP-43 (line 173) acronym is not defined
- Line 212-213: should the reference to Fig 3D and 3E here be interchanged? Figure 3D refers to thin spines but in the text is mentioned in relation to mushroom spine density.
- Figure 6A: WR should be WT?
